# Analysis of German BSE Surveillance Data: Estimation of the Prevalence of Confirmed Cases versus the Number of Infected, but Non-Detected, Cattle to Assess Confidence in Freedom from Infection

**DOI:** 10.3390/ijerph18199966

**Published:** 2021-09-22

**Authors:** Matthias Greiner, Thomas Selhorst, Anne Balkema-Buschmann, Wesley O. Johnson, Christine Müller-Graf, Franz Josef Conraths

**Affiliations:** 1Bundesinstitut für Risikobewertung, 10589 Berlin, Germany; matthias.greiner@bfr.bund.de (M.G.); christine.mueller-graf@bfr.bund.de (C.M.-G.); 2Virtual Centre for Animal Health and Food Safety, University of Veterinary Medicine, 30559 Hannover, Germany; 3Friedrich-Loeffler Institut, 17493 Greifswald-Insel Riems, Germany; anne.balkema-buschmann@fli.de (A.B.-B.); franz.conraths@fli.de (F.J.C.); 4Department of Statistics, University of California at Irvine, Irvine, CA 92697, USA; wjohnson@uci.edu

**Keywords:** BSE, prevalence, disease freedom, non-detects, Germany, Bayesian model

## Abstract

Quantitative risk assessments for Bovine spongiform encephalopathy (BSE) necessitate estimates for key parameters such as the prevalence of infection, the probability of absence of infection in defined birth cohorts, and the numbers of BSE-infected, but non-detected cattle entering the food chain. We estimated three key parameters with adjustment for misclassification using the German BSE surveillance data using a Gompertz model for latent (i.e., unobserved) age-dependent detection probabilities and a Poisson response model for the number of BSE cases for birth cohorts 1999 to 2015. The models were combined in a Bayesian framework. We estimated the median true BSE prevalence between 3.74 and 0.216 cases per 100,000 animals for the birth cohorts 1990 to 2001 and observed a peak for the 1996 birth cohort with a point estimate of 16.41 cases per 100,000 cattle. For birth cohorts ranging from 2002 to 2013, the estimated median prevalence was below one case per 100,000 heads. The calculated confidence in freedom from disease (design prevalence 1 in 100,000) was above 99.5% for the birth cohorts 2002 to 2006. In conclusion, BSE surveillance in the healthy slaughtered cattle chain was extremely sensitive at the time, when BSE repeatedly occurred in Germany (2000–2009), because the entry of BSE-infected cattle into the food chain could virtually be prevented by the extensive surveillance program during these years and until 2015 (estimated non-detected cases/100.000 [95% credible interval] in 2000, 2009, and 2015 are 0.64 [0.5,0.8], 0.05 [0.01,0.14], and 0.19 [0.05,0.61], respectively).

## 1. Introduction

Bovine spongiform encephalopathy was first observed in the United Kingdom in 1986 and spread mainly to other European countries via contaminated cattle feed supplements by trade. The use of rendered bovine carcasses as cattle feed supplements led to exposure of cattle to the prion protein that causes BSE. In 1996, it became evident that a variant of Creutzfeldt-Jakob disease (vCJD) in humans was caused by consumption of beef containing BSE prion protein (for a review, see [1]). This led to massive international concerns regarding the threat that BSE might pose to human health and increased the demand for stricter controls regarding human food of bovine origin.

In the European Union, testing for BSE in cattle is mandatory as per Regulation of the European Parliament and of the Council (EC) No 999/2001 (Annex X, Chapter C). In Germany, BSE testing of all slaughtered cattle above the age of 24, 30, 48, 72, and 96 months was mandatory in 2001–2005, 2006, 2007, 2008–2010, and 2011–2015. Thus, in effect from 28 April 2015, the systematic testing of healthy slaughtered cattle for BSE was no longer mandatory. Nevertheless, testing of risk animals (fallen stock, emergency slaughter) above the age of 48 months has continued to date. Thus, 413 BSE cases were detected out of a total number of 22,039,044 tests conducted until May 2020 [2], at which the query was performed on the HI-Tier database. No further cases have been reported in Germany since 2009 [2]. Testing is performed using approved rapid tests on brainstem samples. The total costs of BSE testing from 2000–2010 in Germany have been estimated between 1847 and 2094 million EUR [3].

In this article, we solely refer to classical BSE as the food-borne type of the disease. Since 2004, low numbers of atypical BSE cases have been reported worldwide, which seem to occur independently from the classical BSE situation [4,5,6,7]. As a sporadic origin can be assumed for these cases, atypical BSE has been excluded from this study.

False-positive test results are considered irrelevant in BSE testing [8,9]. False-negative results, on the other hand, have been mainly attributed to biological characteristics of the infection. Pathogenesis studies suggest that the incubation period and thus indirectly the detection probability depends on various factors such as the infectious dose, route of infection, elapsed proportion of the incubation period, age at infection, and finally on the applied detection system [10,11,12]. One example for accounting for misclassification is from Switzerland, where the observed case rates in cattle have been multiplied by assumed inflation factors ranging between one and four [13]. Similar ratios of non-detected (potentially infectious) cases and detected cases in European cattle have been assumed in the context of quantitative BSE risk assessments [14,15]. More recently, this ratio has been assumed to range between two and ten [14]. Back-calculation has been applied using the empirical cattle survival function and the estimated distribution of the incubation period using data from the BSE epidemic in the UK [16] and, with a similar approach, using data from Japan [9]. Age-specific detection probabilities have been derived from a convolution of probability density functions for the susceptibility and incubation period using data of the UK epidemic [17]. These results have also been used as input into a model for calculating the power of the Danish BSE surveillance system [18]. The power can be interpreted as the confidence in freedom from disease (or from infection) and requires choosing a prevalence threshold (so-called design prevalence), at which at least one case is detected by the surveillance system at a desired power level [19,20,21,22]. To our knowledge, an estimation of the number of BSE infected slaughter cattle that tested negative (“non-detects”) has not been undertaken so far.

Following the distinction between true and apparent prevalence [22], we define for the purpose of our study the true BSE status as relating to the BSE infection status irrespective of the stage of infection and irrespective of the infectivity level of specified risk materials, whereas the apparent BSE prevalence relates to the notified results of the diagnostic tests.

The objectives of our study are (i) to estimate the true birth cohort-specific BSE prevalence in the German cattle population, (ii) to calculate the confidence in freedom from BSE-infection for selected birth cohorts, and (iii) to estimate the number of infected, but non-detected cattle that potentially entered the food chain during the epidemic. All estimates are based on German surveillance data and are adjusted for misclassification.

## 2. Materials and Methods

### 2.1. BSE Surveillance Data

The BSE test data used in the analysis were taken from the German HI-Tier database [23], which contains BSE test results and meta-information, such as time of testing and date of birth of the animal. The considered period spans birth and slaughter cohorts from 1974 to 2020. This period was chosen to include the few animals born from 1974 onwards that are relevant for this analysis. Animals born in and before 1974 were classified into age cohort 1974. Nevertheless, results are reported for birth cohorts 1990 until 2015. We suppose that the probability of escaping from testing, e.g., due to export, is not importantly different between infected and non-infected cattle. The latest birth cohort, in which a classical BSE case was detected, was the 2001 birth cohort (tested in 2005). The analyzed raw data are provided in the supplementary material. The number of tests conducted from 1999 to 2015 are shown in Figure 1.

### 2.2. Statistical Model

*BSE testing results.* We considered as the response variable the number of cattle of birth cohort *i* tested in year *j*, denoted as *n_ij_,* and the corresponding number of positive BSE test results *X_ij_*. The empirical proportion *X_ij_/n_ij_* is the observed prevalence, for convenience expressed as number of observed cases per 100,000 animals.

### 2.3. Age-Dependent Detection Probability

We used the age at testing as a proxy of the incubation period under the assumption that exposure and infection occurs early, i.e., within the first year of life [8,9,13,16,24,25]. Age-dependent detection probabilities (i.e., the diagnostic sensitivity) have been derived from models of the BSE epidemic in the UK [17]. The age-depen-dent detection probability (*δ*) based on UK data can be modelled as a function of age (*a*) using the modified Gompertz function,

(1)
$$\begin{array}{c} \delta \left(a\right)=exp\left(exp\left(\beta_{1}exp\left(\beta_{2}a\right)\right)\right), \end{array}$$

with two parameters (*β*_1_*, β*_2_) (see supplementary material). Here, we define *a* (age) as (0, 1, 2,…) for age groups of 0–12, 13–24, 25–36,..., months. Model (1) accommodates our assumption that the detection probability is close to zero at the time of infection (or at age 0) and that it can reach 100% in BSE-infected cattle that have lived long enough to develop detectable levels of the prion protein. We modified the Gompertz function f(t) = a exp(-b exp(-c t)) by setting the asymptote a = 1 (accounting for the support range of the sensitivity) and omitted the negative sign for the remaining parameters since we obtained more stable results with the MCMC algorithm using negative prior values.

### 2.4. Response Variable

The number of BSE cases from cohort *i* detected in year *j* is modelled as a Poisson (P) variate,

(2)
$$\begin{array}{cc} X_{i,j} & ∼P\left(\lambda_{i,j}\right),\\ \lambda_{i,j} & =n_{i,j}\;\pi_{i}\;\delta \left(a_{i,j}\right) \end{array}$$

where parameter *λ_ij_* is a function of the sample size *n_ij_*, the unknown true cohort-specific BSE prevalence *π_i_* and the age-specific detection probability *δ*(*a*) according to Equation (1). False positive test results are not known to occur in relevant numbers and were therefore not included in the model.

### 2.5. Parameter Estimates

We first considered the true birth cohort-specific BSE prevalence *π**_i_*, estimated from Equation (2) and expressed as cases per 100,000 tested animals as outcome of interest. We then estimated the confidence in freedom from infection [18,21]. This is equal to the probability of observing at least one BSE case in the respective birth cohort with the surveillance system, given a hypothetical threshold prevalence (π^′^; “design prevalence”) of one case out of 100,000 animals in the population. Using an approach described earlier [18], we calculate the contribution of each batch of negative testing results for birth cohort *i* obtained in test year *j*,

(3)
$$\begin{array}{cc} C_{i,j} & =1-\mathrm{Pr}\left(X_{i,j}=0|n_{i,j}\;\pi^{\prime }\;\delta \left(a_{i,j}\right)\right)\\ & =1-{\left(1-\pi^{\prime }\;\delta \left(a_{i,j}\right)\right)}^{n_{i,j}} \end{array}$$

to the cumulative confidence of freedom from BSE infection for birth cohort *i* over all available years of testing, which is given as 
$C_{i}=1-\underset{j}{\prod }\left(1-C_{i,j}\right)$
. The confidence is only applicable to apparently BSE-free birth cohorts.

Finally, we estimated the number of BSE-infected cattle from birth cohort *i* that had been slaughtered and tested negative in year *j* as

(4)
$$\begin{array}{cc} p_{i,j} & =m_{i,j}\;\pi_{i}-m_{i,j}\;\pi_{i}\;\delta \left(a_{i,j}\right)\\ & =m_{i,j}\;\pi_{i}\left(1-\delta \left(a_{i,j}\right)\right), \end{array}$$

where *m_ij_* denotes the number of cattle from cohort *i* slaughtered for human consumption in year *j*. For age groups not subjected to BSE testing, Equation (4) simplifies to *ρ_ij_* = *m_ij_ π*. In the absence of reliable demographic figures, we set *m_ij_* equal to 100,000 and thus quantified the occurrence of non-detects across all birth cohorts as the case rate per 100,000 animals slaughtered for human consumption in year j.

### 2.6. Implementation of the Model

We combined models (Equations (1)–(4)) in a Bayesian framework for simultaneous inference. Four sets of priors for the Gompertz parameters (*β*_1_*, β*_2_) were derived as described in the supplementary material. Briefly, the first set is based on visual inspection of a family of Gompertz curves through coordinates for age group and detection probability (0, 0) and (15, 1) and maximal uncertainty about the point of infection and slope. The second set is the posterior joint distribution obtained from fitting a Gompertz model to age-dependent detection probabilities (i.e., diagnostic sensitivity) used in previous work [18]. A third set of priors has been derived from the experimental results of the German BSE pathogenesis study [26] as described in the Appendix A.

Flat priors for the cohort-specific prevalences *π_i_* were expressed using beta distributions with two shape parameters equal to one. All other estimates are functions of these parameters. We have implemented the model in R [27] using JAGS [28] and the R package R2jags [29]. The point estimates (median values) and the 95% credible intervals were obtained from the joint posterior distribution of the Markov Chain Monte Carlo (MCMC) sample, consisting of 30,000 iterates. Convergence was monitored visually using time series plots of three independent chains and plots of the Gelman-Rubin statistic [30]. The precursor models for generating the Gompertz priors and the final model for inference are provided in the Appendix A.

## 3. Results

We are presenting the results under respective subheadings to provide a concise and precise description of the experimental results, their interpretation, as well as the conclusions that can be drawn.

### 3.1. Birth-Cohort Specific BSE Prevalence Estimates

The prevalence estimates for birth cohorts 1990–2020 of the German cattle population adjusted for misclassification are shown in Figure 1. Among those, the 1996 cohort shows a peak with 16.4 cases and a 95% credible interval (CI) of 13.8–19.1 cases per 100,000 animals (detailed results provided in the Appendix A). The observed BSE rate is below the model-based estimate for all birth cohorts, indicating a systematic underestimation of the true prevalence. The strongest bias occurs for cohort 1999 with an observed proportion of 2.6 per 100,000 compared to a point estimate of 4.33 per 100,000 (95% CI 3.3–5.6). From birth cohort 2005 onwards, the posterior and prior distributions of the prevalence estimates align increasingly, as sample numbers had been continuously reduced under the impression of declining case numbers and a shift of surveillance to more specific risk groups, i.e., fallen stock, culled, and older cattle.

### 3.2. Confidence in Freedom from BSE in Birth Cohorts

No classical BSE cases have so far been found in birth cohorts from 2002 onwards, which qualifies these cohorts as apparently BSE free. We estimated the confidence in the BSE-free status for these apparently free cohorts considering the numbers of tested cattle up to and including test year 2020 (Figure 2a). The confidence level is almost 100% for cohorts 2002–2005 and 99.68% (95% CI 99.6–99.7) for the 2006 cohort. For cohorts of 2007 onwards, the confidence is below the 99% level and shows a declining trend.

### 3.3. BSE Non-Detects

Our model allowed for each year of testing and for each birth cohort estimating the number of BSE-infected cattle that may have entered the food chain, because they tested negative for BSE (“non-detects”) (Figure 2b). In 2001, for example, a testing age of >24 months applied, i.e., all animals that were 24 months old or older had to be tested at slaughter. Thus, birth cohorts 1990–1998 slaughtered in 2001 had to be tested. The estimation of the number of non-detects in these cattle is adjusted for age-dependent misclassification (Equation (4)). On the other hand, cattle from birth cohorts 2000–2001 were not subjected to testing in 2001, and the best estimate of non-detects equals our prevalence estimate (from Equation (2)) for the respective birth cohorts based on all available test results up to and including 2015 (Figure 2b). Our analysis captures the fact that individual cattle from a given birth cohort (e.g., 1999) that reached the test age in the year of slaughter (e.g., 2001) and were subjected to testing, while individual test cohort-mates slaughtered in the same year with an age below the stipulated test age may not have been tested. Both situations were quantitatively evaluated, but cannot be visually differentiated, reflecting the negligible impact of testing the younger age groups. We obtained similar results when we analyzed other testing years. The median number of non-detects per 100,000 animals slaughtered was above 1 per 100,000 in 1990, and 1996 to 1999, and below 1 per 100,000 for the other years of slaughter.

## 4. Discussion

Reliable prevalence estimates are required for assessing the BSE situation from an epidemiological point of view and the potential public health risk posed by BSE-infected cattle. We derived estimates of the true birth cohort-specific BSE prevalence in the German cattle population with adjustment for misclassification. The latter is caused by the long incubation period, slow accumulation of prions, and slow disease progression, and results in the delay of several years after exposure of detecting BSE prions, which has also been shown in experimental studies [26,31].

Our case definition for estimating BSE prevalence is related to infection, regardless of the stage of infection and therefore also regardless of the quantity of infectious BSE agent in edible tissues.

We have stratified prevalence estimates for birth cohorts. This is an important conceptual feature of our model, which reflects the change in hypothetical infection pressure during the course of the BSE epidemic. Epidemiological evidence suggests that exposure of cattle in Germany occurred early during the first year of life [24,25]. Thus, the epidemic curve of prevalence estimates projected upon the year of birth as a proxy for the time of exposure reflects changes of the exposure level over time. For example, the feed ban became effective in January 2001 in Germany, which is consistent with low prevalence figures from this birth cohort onwards.

The estimated epidemic curve projected upon the birth cohorts indicates a strong decline for animals born in 2000 and later, whereby the observed prevalence was zero for all cohorts from 2002 onwards. We estimated that the confidence in freedom from BSE infection is 99% or above for five consecutive birth cohorts from 2002 to 2006. This estimate corresponds to the probability of detecting at least one infected animal with the surveillance system analyzed for a design prevalence of 1/100,000. The latter is a hypothetical concept for quantifying the empirical evidence for freedom from infection when empirical prevalence estimates are zero or not available. Other authors have used a design prevalence for BSE surveillance of 1/10,000 [18] or emphasized that the choice for an appropriate design prevalence is a matter of political and epidemiological judgement [8]. Our choice was motivated by the fact that BSE is not transmissible via direct contact between cattle and thus may occur at a very low prevalence in an affected population.

Our estimates of the number of non-detects are based on the adjustment for misclassification. The misclassification model suggests that mainly younger infected cattle are prone to false-negative testing results. However, the BSE agent is unlikely to be present in infectious concentrations in edible tissues of younger animals, as experimental studies have shown [26,32]. In particular, the removal of specified risk materials (SRM) mitigates the potential risk for human infection effectively. Therefore, we conclude that non-detects that may have entered the food chain are no major concern from a consumer safety point of view.

The estimates of the cohort-specific true prevalence and the non-detects increase for the 2016–2020 birth cohorts, while the estimated probability of freedom from disease decreased for the 2016–2020 birth cohorts. To understand this, it is important to point out that the estimated true prevalence is used in the calculation of these parameters. When considering the distributions of estimated true prevalences for the birth cohorts from 2006 onward, it is obvious that the distribution is uninformative and corresponds exactly to the prior distribution. This effect occurs, since there is not enough empirical information (i.e., test data) in the observations for the 2016–2020 birth cohorts to override the prior distributions (i.e., the prior assumptions about the true prevalence). In the model chosen here, the prior distributions for cohort-specific prevalences were non-informative beta(1,1) distributions, which resemble a uniform distribution in the interval [0,1]. These priors are conservative (i.e., higher values of central tendency than plausible in reality) compared to epidemiological reasoning about the occurrence of BSE in the respective birth cohorts. We chose this conservative approach to avoid any underestimation of true prevalence in birth-cohorts for which surveillance data are available. This evidence-based approach precludes any inferences about data-sparse birth cohorts 2016–2020.

The increase suggested by the model is an effect of the method: If there is not enough data (information), the model ‘sticks’ to the *a priori* assumptions (uniform distribution in the interval [0,1] with median 0.5 or 50,000 positives in 100,000 animals).

It was known from other sources of evidence (experimental data, decreasing case numbers in slaughtered cattle in Germany and elsewhere, etc.) that there had been an almost complete disruption of the infection events after the “feed ban” had been enforced. Based on this knowledge, the surveillance measures were adjusted over time to test older animals and fallen stock, so that the testing of healthy younger cattle at slaughter could be reduced. This led to the decrease in confidence for the surveillance compound of testing healthy young cattle at slaughter, which was compensated, however, by testing all fallen stock, culled and older cattle, so that the total sensitivity and reliability of the BSE surveillance system was not impaired.

## 5. Conclusions

In this analysis of the German BSE epidemic, we calculated birth cohort-specific classical BSE prevalence estimates for the German cattle population adjusted for diagnostic misclassification using a Bayesian model. The model allows estimating the age-depending detection probability, which can be interpreted as the diagnostic sensitivity, from the surveillance data. Birth cohort-specific BSE prevalence estimates can be used to project the BSE epidemic curve on the birth cohorts. Under the realistic assumption of exposure to and infection with BSE prions early in life, this projection is an appropriate reflection of the changes in the infection pressure over time. The results of our model also suggest that the birth cohorts from 2002–2006 can be assumed as free from classical BSE using a formal quantification of the evidence from available negative testing results. The number of BSE-infected, but non-detected cattle that may have entered the food chain as estimated by our model is a minor, if not negligible, public health risk. Any potentially remaining public risk was further reduced by other preventive measures (e.g., testing all fallen stock, culled and older cattle, removing specified risk material, etc.), some of which are still in place.

## Figures and Tables

**Figure 1 ijerph-18-09966-f001:**
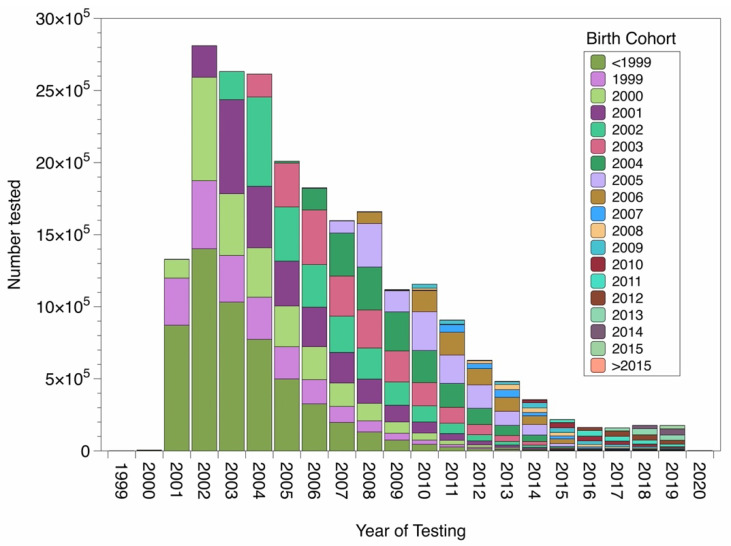
Slaughter animals tested for BSE stratified by birth cohort.

**Figure 2 ijerph-18-09966-f002:**
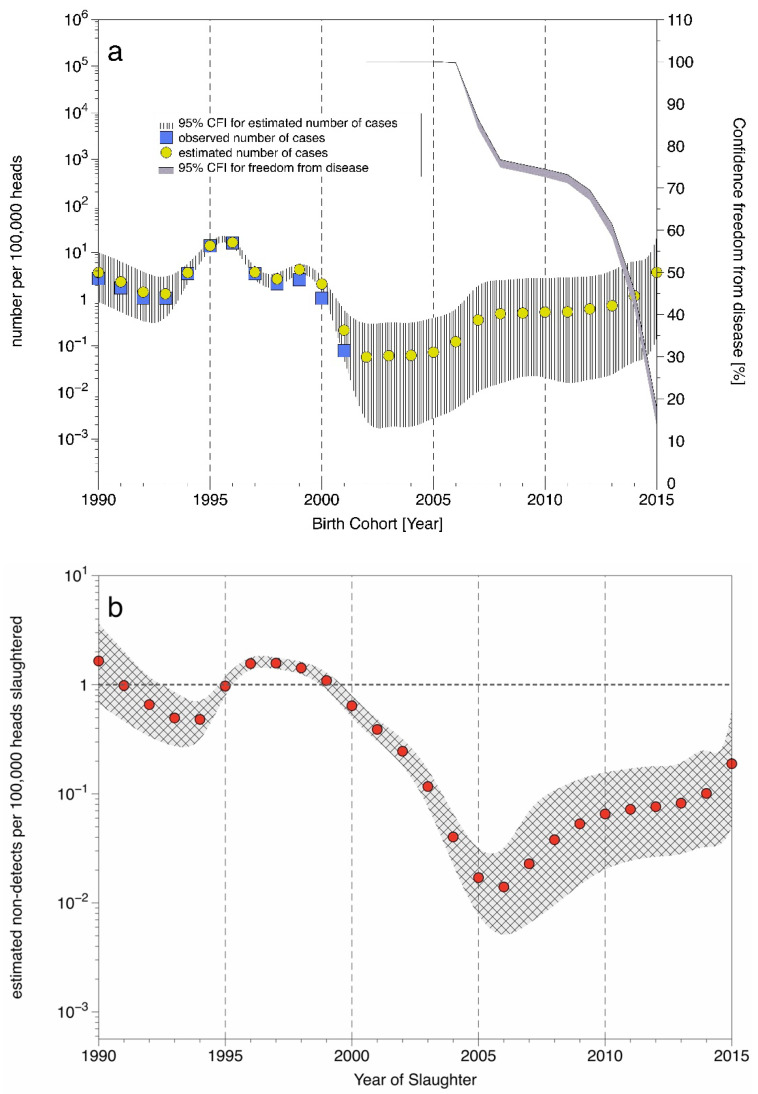
(**a**) Observed apparent BSE prevalence (per 100,000 heads slaughtered), model-based estimated BSE prevalence adjusted for misclassification, and estimated confidence of freedom from disease [%] per birth cohort. (**b**) Estimated number of BSE infected cattle per 100,000 not detected by the surveillance system and thus entering the food chain. 95% credibility intervals are shown for the estimates (hatched area).

## Data Availability

The data presented in this study are available on request from the corresponding author. The data are not publicly available due to privacy issues.

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
