# Peer review of "Analysis of German BSE Surveillance Data: Estimation of the Prevalence of Confirmed Cases versus the Number of Infected, but Non-Detected, Cattle to Assess Confidence in Freedom from Infection"

_ijerph, 2021, doi:10.3390/ijerph18199966_

Round 1

Reviewer 1 Report

This study aimed to build statistical models to quantitatively assess the risk of Bovine spongiform encephalopathy (BSE), including the prevalence of infection, the probability of absence of infection in defined birth cohorts and the number of infected but non-detected, based on German BSE surveillance data. The questions the study tried to solve are interesting and important from the public health perspective. However, there are several points needed to clarify or improve, as following:

(1) This study estimated the number of infected but non-detected mainly based on introducing an age-specific detection probability (δ) for diagnostic sensitivity. However, sensitivity may influence by many other factors. Please clarify that why the authors just considered this age-specific detection probability and ignored other factors? Would such way lead to bias?

(2) For each cohort, just one true prevalence was set, as the authors assumed that “exposure and infection occurs early, i.e. within the first year of life”. Would this assumption lead to any bias? Is it possible that cattle could be infected latter in their lives and the prevalence of each cohort change during the lifetime?

(3) The definition for “confidence in freedom from infection” (lines 119-123) seems obscure. Why “observing at least one BSE case” out of 100,000 animals could equal to “freedom from infection”, but not “at most one BSE case”? After all, freedom from infection should be equal or less than the threshold instead of larger than that. Moreover, the authors used formula (3) in the manuscript to estimate the confidence in freedom from infection. However, it seems contradictory with the prevalence, as π increases, the C will increase, which means that increasing the prevalence will lead to larger the confidence in freedom from infection. Please clarify the above two contradictions.

(4) The authors did model verification using a synthetic data set. However, how was the model fitting and prediction regards to the true data seems not provided. Could the authors provide such results?

(5) For processing the surveillance data, the authors stated that they adjusted the misclassification. What does the “misclassification” mean? Please clarify it.

(6) In Figure 2, the estimation of numbers of non-detected seems larger than the numbers of all cases in the year 2001-2003, as the yellow points below the red points during these years, which seems unreasonable. How to explain it?

(8) Please write the full name of BSE in the title. Besides, there are several typing errors in the manuscript, such as “0,216” in line 20, “f BSE” in line 206. Please carefully check and avoid such errors.

(9) In the abstract, it would be better to report results with credible interval instead of just point estimates. Besides, the conclusion in the abstract seems not be supported by the results. Please make more reasonable conclusion.

(10) In the introduction, please added contents on the public health importance of the disease.

Reviewer 2 Report

Dear authors,

this manuscript introduce and interest topic related to the German surveillance data, because you face a low prevalence phenomenon by means of an appropriate bayessian approach arising from a Poisson model due to this low prevalence. The manuscript is well written from a friendly reading point of view. It's a success not to introduce the heavy mathematical and statistical foundation within the text but in a supplementary material because it makes readable your manuscript for non-statisticians.

I would like to highlight the quality of your explanations as well as you provide answers to possible concerns before the question is made. For instance, I was thinking in my first review that a non-informative Beta as prior distribution was not going to be appropriate, but you justify this fact along the text and I understand pefectly why this is a good choice.

However, I have some minor questions related to the manuscript and the supplementary material that you could take into account. 

Regarding to the manuscript:

  • Review the numbers of all equations because they are duplicated.
  • Please use the classical P instead of Poi to denote the Poisson distribution. 
  • You introduce the age-dependent detection probability function in lines 102-109. Here you should guarantee that it is a real probability function, I mean: the values are between 0 and 1. I think that it is correct because of the values of the two parameters but it is important to highlight that due to this fact is well defined as a probability function. Here you use the modified Gompertz function but, as you know, since the first definition of Benjamin Gompertz there has been many modifications of this function to be applied in biology, among others. So, in my opinion, it would be interesting to introduce the reference of this modification that you are using or to highlight that it is an own proposal of modified Gompertz function.

Regarding to the suplementary material:

  • Review the sentence <<... Gompertz ffuntion isrequiered...>>
  • I think that once you provide your R source code, this is very worthy, you could to highlight in text usin for example a box for this sentences.
  • Review the american or english spelling to use one of them. I mean that you seem to use english spelling but you write, for instance, 'analyzed' in the suplementary material.

Nothing more, if the authors are so kind to consider some of my comments the paper will be ready for publication.
